# Skyrmion lattice in centrosymmetric magnets with local Dzyaloshinsky-Moriya interaction

Shi-Zeng Lin[1]

[1]*Theoretical Division, T-4 and CNLS, Los Alamos National Laboratory,*

*Los Alamos, New Mexico 87545, USA*

(Dated: December 23, 2021)

## Abstract

It is common that the local inversion symmetry in crystals is broken, even though the whole crystal has global inversion symmetry. This local inversion symmetry breaking allows for a local Dzyaloshinsky-Moriya interaction (DMI) in magnetic crystals. Here we show that the local DMI can stabilize a skyrmion as a metastable excitation or as a skyrmion crystal in equilibrium. We consider crystal structure with layered structure as an example, where local inversion is violated in each layer but a global inversion center exists in the middle of the two layers. These skyrmions come in pairs that are related by the inversion symmetry. The two skyrmions with opposite helicity in a pair form a bound state. We study the properties of a skyrmion pair in the ferromagnetic background and determine the equilibrium phase diagram, where a robust lattice of skyrmion pairs is stabilized. Our results point to a new direction to search for the skyrmion lattice in centrosymmetric magnets.

## I. INTRODUCTION

Magnetic skyrmions are topologically protected localized excitations, which have attracted considerable attention recently. [1–4] Skyrmions have lots of promising applications in spintronic devices because of their compact size, high mobility and stability. Both single skyrmion and skyrmion lattice have been observed in a wide classes of magnetic materials indicating their ubiquitous existence. This includes chiral magnets [5–8], magnetic multilayers [9, 10] and centrosymmetric magnets [11–13]. In chiral magnets and multilayers, the skyrmions are stabilized by the Dzyaloshinsky-Moriya interaction (DMI) [14–16] as a consequence of the inversion symmetry breaking. [17–19] While in the centrosymmetric magnets, skyrmions are stabilized by the frustrated or competing magnetic interactions. [20–22] Despite the tremendous progress in the past in identifying new skyrmion hosting materials, it is always demanding to find a new mechanism to stabilize skyrmions, as such a new mechanism will likely lead to novel physical properties of skyrmions, which may be desirable for device applications.

Here we demonstrate that the local DMI in globally centrosymmetric magnets can also support skyrmions. The global centrosymmetry alone is not enough to forbid DMI. Instead, the inversion symmetry can be broken locally which admits a local spin orbit coupling and hence DMI. One example is the spins in honeycomb lattice. Two nearest neighbor spins are inversion symmetric with the inversion center localized at the center of the bond, thus forbids the DMI. However, there is no inversion symmetry for the two next nearest neighbor spins, and therefore a local DMI is allowed. The honeycomb lattice has the mirror symmetry with respect to the honeycomb plane. As a result, the Moriya rule [15] dictates the DMI vector being perpendicular to the honeycomb plane. Including such a DMI to the nearest neighbor ferromagnetic Heisenberg model on the honeycomb lattice realizes a Haldane model for the magnons with topological magnon bands. [23]

## II. MICROSCOPIC MODEL

To be specific, let us consider crystal structure of type $CaBe_2Ge_2$ (space group 129, see Fig. 1 for a schematic view [24]). The magnetic ions (light orange) form layered square lattice with odd and even layer being related by the inversion symmetry. Inside each layer, the local inversion symmetry is broken, which allows for a Rashba spin orbit interaction. The local Hamiltonian for

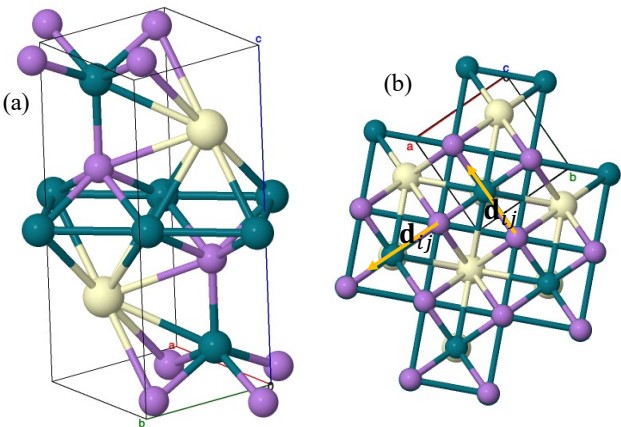

FIG. 1. (a) Unit cell of the crystal structure of type $CaBe_2Ge_2$. We consider a material system with two magnetic ions (light orange sphere) in a unit cell. Each magnetic ion is not an inversion center, but the two magnetic ions are related by inversion with the inversion center located at the middle of the bond connecting these two ions. (b) Enlarged unit cell where magnetic ions form a layered square lattice. Within each layer, the local inversion symmetry is broken, but a pair of the nearest neighboring two layers enjoy the inversion symmetry. The golden arrows represent the spin orbit coupling vectors.

each layer is assumed to be

$$\mathscr{H}_l = -t \sum_{\langle i,j \rangle, \alpha} c_{il\alpha}^\dagger c_{jl\alpha} - i\lambda \sum_{\langle i,j \rangle, \alpha\beta} c_{il\alpha}^\dagger \sigma_{\alpha\beta} \cdot (-1)^l \mathbf{d}_{ij} c_{jl\beta} + \text{H.C.} - \frac{J_H}{2} \sum_{i,\alpha\beta} c_{il\alpha}^\dagger \sigma_{\alpha\beta} c_{il\beta} \cdot \mathbf{S}_{il}. \quad (1)$$

Here $i$, $j$ are site indices of the square lattice at each layer $l$. We consider nearest neighboring hopping between electrons described by the creation operator $c_{il\alpha}^\dagger$ with spin index $\alpha$. We have assumed a local exchange coupling between classical spin $\mathbf{S}_{il}$ with $|\mathbf{S}_{il}| = 1$ and the electron spin. $\lambda$ is the strength of the Rashba spin orbit interaction characterized by a unit vector $(-1)^l \mathbf{d}_{ij}$ that reverses its direction from layer to layer. The unit vector is perpendicular to the bond for the Rashba spin orbit interaction $\mathbf{d}_{ij} = \hat{z} \times \mathbf{r}_{ij}/|r_{ij}|$ with $\mathbf{r}_{ij} = \mathbf{r}_i - \mathbf{r}_j$. We assume a hopping between electrons in the nearest layers only when the site is aligned and neglect the interlayer spin orbit coupling (the interlayer spin orbit coupling is allowed even there exists an inversion center),

$$\mathscr{H}_c = -t_c \sum_{i,l} c_{il+1\alpha}^\dagger c_{il\alpha} + \text{H.C.}. \quad (2)$$

The total Hamiltonian $\mathscr{H}_T = \mathscr{H}_c + \sum_l \mathscr{H}_l$ is invariant under inversion transformation, which flips the even and odd layers, i.e. $l \leftrightarrow l+1$. The coupling of localized spin $\mathbf{S}_{il}$ to the conduction

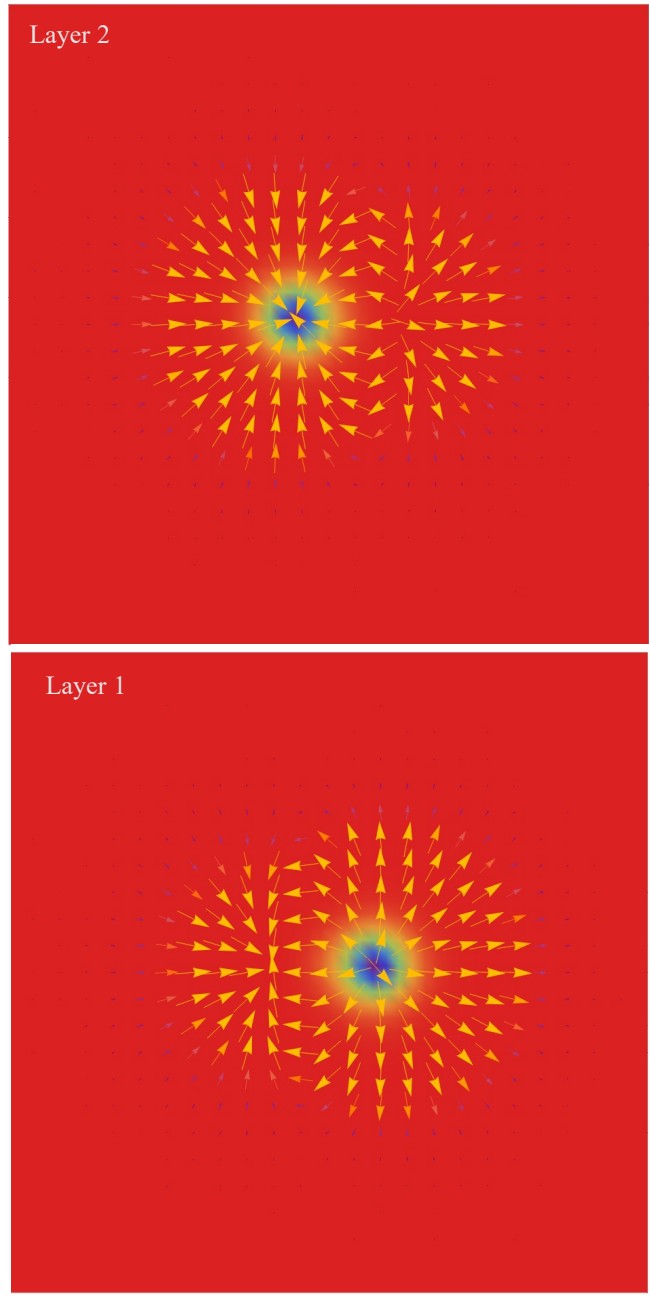

FIG. 2. Spin configuration of a skyrmion pair. Color represents out-of-plane component of spin (blue: -1 and red: +1) and arrows denote the in-plane component. There is a relative shift between the two skyrmions in a skyrmion pair. Meanwhile, a skyrmion in one layer induced a halo in its neighboring layers. Here $B = 1.0$ and $J_{12} = 0.4$.

electrons mediates an effective magnetic interaction between $\mathbf{S}_{il}$. In the strong coupling limit $J_H \gg t$, $t_c$ (double-exchange mechanism), we can consider two site problem with $i = 1, 2$. [25] For $\mathbf{S}_{il}$ in the same layer, by performing SU(2) rotation in the spin space, i.e. $\tilde{c}_{1l\alpha} = [\exp(-i\theta\boldsymbol{\sigma} \cdot (-1)^l \mathbf{d}_{12})/2]_{\alpha\beta} c_{1l\beta}$ and $\tilde{c}_{2l\alpha} = [\exp(i\theta\boldsymbol{\sigma} \cdot (-1)^l \mathbf{d}_{12})/2]_{\alpha\beta} c_{2l\beta}$ with $\tan\theta = \lambda/t$, the spin orbit coupling can be absorbed into an effective hopping parameter. The localized moment after the rotation is given by

$$\tilde{\mathbf{S}}_{1l} = \cos\theta \mathbf{S}_1 - \sin\theta(\mathbf{S}_{1l} \times (-1)^l \mathbf{d}_{12}) + (1 - \cos\theta)(\mathbf{S}_{1l} \cdot \mathbf{d}_{12}))\mathbf{d}_{12})). \tag{3}$$

For $\tilde{\mathbf{S}}_{2l}$, we need to replace $\theta \to -\theta$ in the above equation. After the SU(2) rotation, the Hamiltonian is the same as the Anderson-Hasegawa Hamiltonian without a spin orbit coupling [26], and then we obtain the interaction between $\tilde{\mathbf{S}}_{il}$, $\mathscr{H} = -\kappa\tilde{t}\sqrt{1 + \tilde{\mathbf{S}}_{1l} \cdot \tilde{\mathbf{S}}_{2l}/2}$, where $\tilde{t} \equiv \sqrt{t^2 + \lambda^2}$ and $\kappa$ is a constant depending on the electron density. In term of the original classical spin $\mathbf{S}_{il}$, the Hamiltonian becomes

$$\mathscr{H} = -J\sum_{\langle i,j\rangle, l} \mathbf{S}_{il} \cdot \mathbf{S}_{jl} - J_{12}\sum_{i,l} \mathbf{S}_{il+1} \cdot \mathbf{S}_{il} - \sum_{\langle i,j\rangle, l} \mathbf{D}_{ij} \cdot \mathbf{S}_{il} \times \mathbf{S}_{jl} - A\sum_{i,l}(S_{il}^x S_{i+\hat{y}l}^x + S_{il}^y S_{i+\hat{x}l}^y). \tag{4}$$

Here $J = \tilde{t}\kappa\cos(2\theta)$, the compass anisotropy strength $A = \tilde{t}\kappa(1 - \cos(2\theta))$ and the interlayer coupling strength $J_{12} = t_c\kappa$. $\hat{x}$, $\hat{y}$, $\hat{z}$ are unit vector in the $x$, $y$, $z$ direction respectively. The DMI vector is perpendicular to the bond with $\mathbf{D}_{i,i+\hat{x}} = \tilde{t}\kappa\sin\theta\hat{y}$ and $\mathbf{D}_{i,i+\hat{y}} = -\tilde{t}\kappa\sin\theta\hat{x}$ (see Fig. 1). For realistic material parameters, $\lambda \ll t$, we have $J \gg |D_{ij}| \gg A$. The DMI is nonperturbative because any weak DMI turns the ferromagnetic (FM) state into a magnetic spiral state, hence it should not be neglected even though it is weak. In the following discussion, we will neglect the $A$ term. The same form of $\mathscr{H}$ can also be derived for the superexchange mechanism [25], which is at work in Mott insulators. For RKKY interaction valid in the region $J_H \ll t$, there appears long range interaction between localized moments. In view of the broad relevance of the Hamiltonian Eq. (4), we will treat $J_{12}$ as a free parameter which can either be antiferromagnetic (AFM) or FM. For $|\mathbf{D}_{ij}|/J \ll 1$, the relevant length scale of the possible spin textures is much longer than the atomic (or tight-binding) lattice parameter. We can write Eq. (4) in the continuum limit

$$\mathscr{H} = \sum_l \int dr^2 \left[\frac{J}{2}(\nabla \mathbf{S}_l)^2 + D(-1)^l [S_z(\nabla \cdot \mathbf{S}) - (\mathbf{S} \cdot \nabla)S_z] - J_{12}\mathbf{S}_l \cdot \mathbf{S}_{l+1} - \mathbf{B} \cdot \mathbf{S}_l\right], \tag{5}$$

where we have included a Zeeman coupling to spins and $D$ is the magnitude of DMI. When $J_{12} = 0$, Eq. (5) reduces two copies of well studied Hamiltonian for chiral magnets, which supports both single skyrmion and skyrmion lattice solutions. The spin texture depends on the form of the DMI.

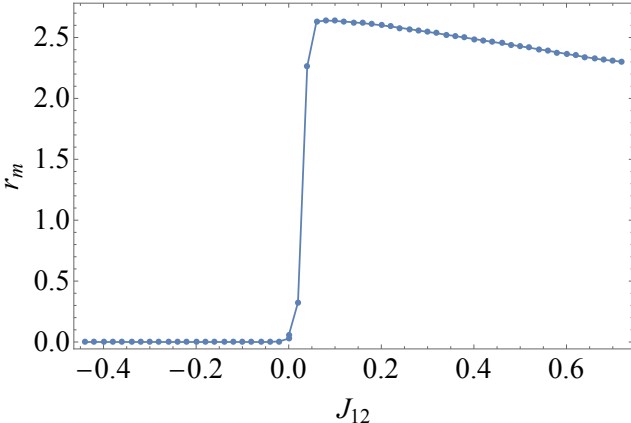

FIG. 3. Optimal relative separation between the two skyrmions in a skyrmion pair obtained by numerical minimizing of the energy. The skyrmion pair becomes unstable when $J_{12}$ is outside of the range plotted in the figure. Here $B = 1.0$.

In our case, the skyrmion is of Neél type. In one layer, the in-plane component of the spins of a skyrmion are pointing outward along the radial direction of the skyrmion while in the other layer, the spins are pointing inward, see Fig. 2 for an example. The former skyrmion has helicity 0 and the latter has helicity $\pi$. Such spin configuration costs energy in interlayer coupling $J_{12}$, no matter if it is AFM or FM. Our task is to understand the behavior of single skyrmion metastable state in the ferromagnetic background and the skyrmion lattice in the presence of a nonzero $J_{12}$. In the following, we will take the two dimensional limit by keeping minimal two layers in Eq. (5). We will use dimensionless unit by normalizing length in unit of $J/D$ and energy in unit of $J^2/D$. The model has two independent parameter $B$ and $J_{12}$, both of which are in unit of $D^2/J$.

## III. SINGLE SKYRMION SOLUTION

Let us first study the properties and dynamics of a skyrmion pair in the ferromagnetic background, with one skyrmion sitting in the top layer and the other skyrmion sitting in the bottom layer. The top and bottom layers are related by the inversion symmetry. The spins away from the skyrmion center are fully polarized by the magnetic field, which requires $B$ being much larger than $|J_{12}|$ if $J_{12} < 0$. We minimize the energy numerically with a skyrmion pair as an initial condition. The two skyrmions in the skyrmion pair form a bound state with attraction between the two skyrmions. The relative shift of the two skyrmions depends on $J_{12}$. For $J_{12} < 0$ (AFM interlayer

coupling), the two skyrmions in different layers sit on top of each other. While for $J_{12} > 0$ (FM interlayer coupling), there is a relative shift between the two skyrmions and the relative shift increases with $|J_{12}|$, see Fig. 3. When there is a relative shift, the skyrmion in on layer couples to the otherwise ferromagnetic state in the neighboring layer and generates a shadow spin texture with winding of spins, as displayed in Fig. 2. The reason why the two skyrmions in two layer shifts from each other for $J_{12} > 0$ can be understood as follows. Due to the opposite helicity of the two skyrmions, the in-plane component of the spins associated with the two skyrmions are opposite in the two layers, which is favored by $J_{12} < 0$. For $J_{12} > 0$, the two skyrmions avoid siting atop of each other to save cost in the ferromagnetic coupling between the in-plane component of spins. The out-of-plane component of spins are less important in determining the relative position of skyrmions because the Zeeman coupling dominates over other interactions.

To quantify the shape of the skyrmion pair, we introduce the skyrmion topological charge center for a skyrmion in the $l$-th layer

$$\mathbf{r}_l = \int_{\text{l-th layer}} dr^2 \mathbf{r}_l \mathbf{S}_l \cdot \partial_x \mathbf{S}_l \times \partial_y \mathbf{S}_l \Big/ \int_{\text{l-th layer}} dr^2 \mathbf{S}_l \cdot \partial_x \mathbf{S}_l \times \partial_y \mathbf{S}_l \tag{6}$$

and the relative shift $r_{12} = |\mathbf{r}_1 - \mathbf{r}_2|$. The optimal $r_{12}$ (denoted as $r_m$) that minimizes the skyrmion pair energy as a function of $J_{12}$ is shown in Fig. 3. The separation between the two skyrmions quickly saturates to a value and then decreases slowly as $J_{12}$ is increased. The skyrmion pair becomes unstable when $J_{12} > 0.72$ or $J_{12} < -0.44$ at $B = 1.0$.

Now we turn the dynamics of the skyrmion pair. We consider both the spin transfer torque and spin Hall torque. The dynamics of spin is given by

$$\partial_t \mathbf{S} = -\gamma \mathbf{S} \times \mathbf{H}_{\text{eff}} + \alpha \mathbf{S} \times \partial_t \mathbf{S} + \tau, \tag{7}$$

where $\mathbf{H}_{\text{eff}} \equiv -\delta \mathcal{H}/\delta \mathbf{S}$ is an effective field, $\alpha$ is the Gilbert damping and $\gamma$ is the gyromagnetic ratio. We use $\alpha = 0.2$ in simulations. The torque is $\tau = \frac{\hbar\gamma}{2e}(\mathbf{J} \cdot \nabla)\mathbf{S}$ for the spin transfer torque [27, 28] and $\tau = \frac{\hbar\gamma\theta_{\text{sh}}}{2ed}\mathbf{S} \times [\mathbf{S} \times (\hat{z} \times \mathbf{J})]$ for the spin Hall torque [29–31]. Here $\theta_{\text{sh}}$ is the spin Hall angle, $d$ is the film thickness and $e > 0$ is the elementary electric charge. For a weak drive, the deformation of skyrmion is negligible [32] and the skyrmion dynamics are described by its translational motion in space, which is a Goldstone mode in a clean system. The translational motion is parameterized by the skyrmion center $\mathbf{R}_l$ and the corresponding velocity $\mathbf{v}_l = \dot{\mathbf{R}}_l$, which obey the Thiele's equation [33]. For the spin transfer torque, the Thiele's equation for the two

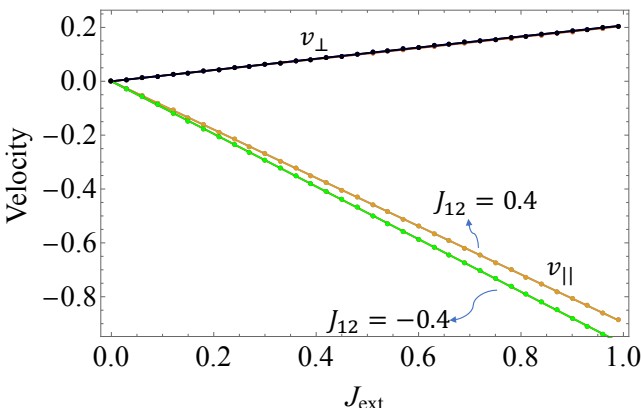

FIG. 4. Skyrmion dynamics under a spin transfer torque. The two skyrmions in a skyrmion pair move as a whole, and move almost anti-parallel to the applied current direction with a small transverse velocity proportional to the damping constant. The velocity vs $J_{ext}$ for $J_{12} < 0$ is the same as that for $J_{12} = 0$ within numerical resolution. The velocity at a given $J_{ext}$ becomes smaller when $J_{12} > 0$ is increased due to the induced shadow texture in the neighboring layer. Here $B = 1.0$ and time is in unit of $J/(\gamma D^2)$.

skyrmions in a skyrmion pair reads

$$Q_l \hat{z} \times \left( \mathbf{v}_l + \frac{\hbar \gamma}{2e} \mathbf{J}_{ext} \right) = -\alpha \eta_l \mathbf{v}_l + (-1)^l \mathbf{F}_{12}, \tag{8}$$

where $\mathbf{F}_{12}$ is the attraction between the two skyrmions at two layers. Here $\eta_l$ is the skyrmion form factor $\eta_l \equiv \int'_{l-\text{th layer}} (\partial_\mu \mathbf{S}_l)^2 dr^2/(4\pi)$ and the skyrmion topological charge $Q_l = \int'_{l-\text{th layer}} dr^2 \mathbf{S}_l \cdot \partial_x \mathbf{S}_l \times \partial_y \mathbf{S}_l/(4\pi)$, where the integration is restricted to the region around a skyrmion. [34] The Thiele's equation for the spin transfer torque does not depend on the helicity of the skyrmions and hence the two skyrmions respond to the spin transfer torque in the same way. The two skyrmions move as a whole skyrmion pair almost the same as those in the decoupled limit $J_{12} = 0$. The skyrmion pair velocity can be found $v_\parallel = -\gamma \hbar Q^2 J_{ext}/[2e(Q^2 + \alpha^2 \eta^2)]$ and $v_\perp = -\gamma \hbar \alpha \eta Q J_{ext}/[2e(Q^2 + \alpha^2 \eta^2)]$, where $v_\parallel$ ($v_\perp$) is the skyrmion velocity component parallel (perpendicular) to the current. We have approximated $\eta = \eta_l$ and $Q = Q_l$. The numerical results are shown in Fig. 4. The interlayer coupling $J_{12}$ enters into the equation through the form factor $\eta_l$ because $J_{12}$ causes skyrmion deformation (see Fig. 2). For $J_{12} > 0$, there are a relative shift between the two skyrmions and induced shadow in the neighboring layers. As a consequence $\eta_l$ for $J_{12} > 0$ is larger than these for $J_{12} \leq 0$. This explains the result in Fig. 4, where $|v_\parallel|$ for $J_{12} = 0.4$ is smaller than that for $J_{12} = -0.4$.

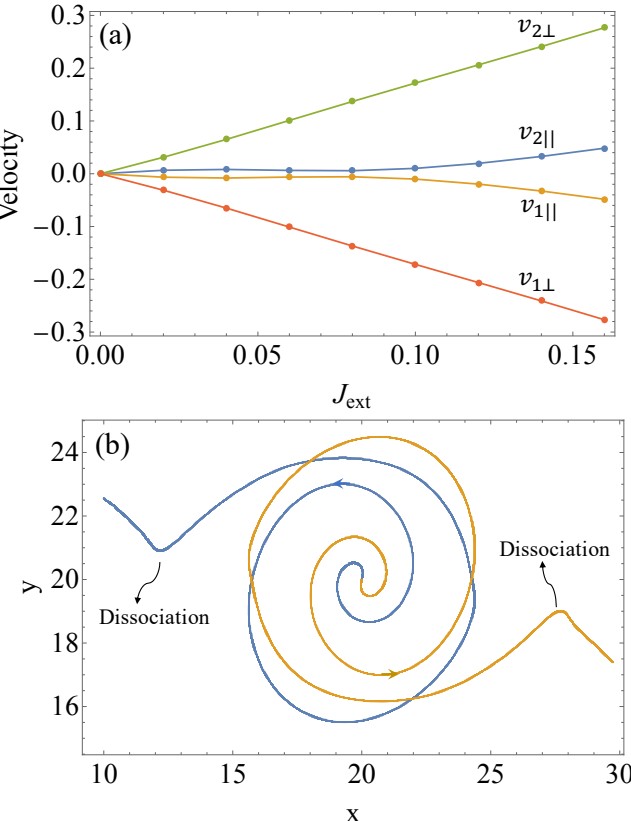

FIG. 5. (a) Velocity of the two skyrmions after a skyrmion pair disassociates at the threshold current $J_{\text{ext}}^c = 0.0085$ for $B = 1.0$ and $J_{12} = -0.4$. The two skyrmions travel in the opposite direction. (b) Trajectory of the two skyrmions in a skyrmion pair subjected to a spin Hall torque. Initially the two skyrmions are sitting atop of each other for $J_{12} = -0.4$. The point when the skyrmion pair disassociates is indicted in the figure.

The dynamics of skyrmion pair driven by the spin Hall torque is more interesting because the spin Hall torque couples to the helicity. [35] The Thiele's equation in this case is

$$Q_l \hat{z} \times \mathbf{v}_l + \alpha \eta_l \mathbf{v}_l = \frac{\hbar \gamma \theta_{\text{sh}}}{2ed} J_{\text{ext}} \mathbf{Y}_l + (-1)^l \mathbf{F}_{12}, \tag{9}$$

$Y_{l,\mu} = (\hat{z} \times \hat{\mathbf{J}}_{\text{ext}}) \cdot \int_{l-\text{th layer}}' (\partial_\mu \mathbf{S}_l \times \mathbf{S}_l) \, dr^2/(4\pi)$ with $\mu = x, y$. $\hat{\mathbf{J}}_{\text{ext}}$ is a unit vector along the current direction. It is straightforward to verify $Y_{1,\mu} = -Y_{2,\mu}$ when the two skyrmions have opposite helicity. Therefore the two skyrmions in a skyrmion pair tend to move in a different trajectory. For Néel skyrmions with helicity equals 0 or $\pi$, $\mathbf{Y}_l$ is parallel to the current direction and its magnitude is denoted by $Y_{l\parallel}$. When the current is small, the force $F_{12}$ balances the different driving force acting on the two skyrmions by adjusting the relative shift of position between the

two skyrmions. The corresponding solution is $\mathbf{v}_l = 0$ and $|F_{12}| = |\frac{\hbar\gamma\theta_{sh}}{2ed}J_{ext}\mathbf{Y}_l|$, When the current reaches a threshold, the maximal force $F_{12}$ between the two skyrmions is no longer sufficient to cancel the disparity in the driving force between the two skyrmions. As a result, the skyrmion pair disassociates and the two skyrmions move independent. One can obtain the skyrmion velocities by setting $F_{12} = 0$ in Eq. (9), which yields $v_{1\perp} = -QY_{1\parallel}\hbar\gamma\theta_{sh}J_{ext}/[2ed(Q^2 + (\alpha\eta)^2)]$ and $v_{1\parallel} = \alpha\eta Y_{1\parallel}\hbar\gamma\theta_{sh}J_{ext}/[2ed(Q^2 + (\alpha\eta)^2)]$, and similarly for $\mathbf{v}_2$. Because $Y_{1,\mu} = -Y_{2,\mu}$, the two skyrmions move in the opposite direction, see Fig. 5 (a). The skyrmions travel almost perpendicularly to the current direction, $v_{1\perp} \gg v_{1\parallel}$.

Let us examine the dynamical process of the decoupling of the two skyrmions in a skyrmion pair. We take $J_{12} < 0$ such that two skyrmions sit on top of each other. We then turn on current and calculate $\mathbf{r}_l$. The trajectory is shown in Fig. 5 (b). As current is turned on, the two skyrmions move away from each other. The trajectory is a spiral because of the Magnus force, $\sim \hat{z} \times \mathbf{v}_l$, which forces skyrmion to move perpendicularly to the force direction. When the skyrmion separation exceeds a threshold value, the two skyrmions decouple.

Interestingly, one can estimate $F_{12}(r)$ and hence the potential $U(r)$ from Eq. (9). This can be done in two different ways. In the first approach, one considers the static limit for a small current. Then $F_{12}(r_m) = |\frac{\hbar\gamma\theta_{sh}}{2ed}J_{ext}\mathbf{Y}_l|$ where $r_m$ is the equilibrium relative distance between the two skyrmions. In the second approach, one directly calculates $F_{12}(r)$ from the trajectory of two skyrmions upon switching on current.

It is also interesting to ask if the internal force $F_{12}$ produces an effective mass for the skyrmion pair, which causes retarded response to the external drive. The answer is no. This can be seen from Eqs. (8) or (9) by rewriting the equation of motion in term of relative coordinate $\mathbf{R}_{12} = \mathbf{R}_1 - \mathbf{R}_2$ and center of mass coordinate $\mathbf{R} = \mathbf{R}_1 + \mathbf{R}_2$. No term of the form $M\ddot{\mathbf{R}} + \cdots = \mathbf{F}_{ext}$ with $\mathbf{F}_{ext}$ being a driving force, is generated, hence the mass generation due to $\mathbf{F}_{12}$ is absent. However, the skyrmion mass can be generated through excitation of high energy internal modes [32, 36, 37], which has been neglected here.

## IV.   SKYRMION LATTICE

We then determine the equilibrium phase diagram of Hamiltonian Eq. (5) in two dimensions. We perform numerical annealing from high temperature disordered state to zero temperature and determine the ground state spin configuration. We discretize Eq. (5) into a square lattice with a

lattice parameter $dr = 0.8$, such that the magnetic spiral at $B = 0$ and $J_{12} = 0$ roughly has 8 lattice parameters. The phase diagram is displayed in Fig. 6 where the triangular lattice of skyrmion pair is stabilized in a wide region in the $B$ and $J_{12}$ space. A large value of $|J_{12}|$ disfavors the skyrmion crystal. When it is stabilized, the skyrmion crystal is more stable for an AFM $J_{12}$ than that for a FM $J_{12}$. The skyrmion crystal is aligned in the direction perpendicular to the layers when $J_{12} < 0$. While for $J_{12} > 0$, the in-plane position of the skyrmions locates at positions corresponding to the interstitial sites of the skyrmion lattice in the nearest neighboring layers. As such relative shift is not always optimal for a skyrmion pair (see Fig. 2), this explains why the skyrmion crystal is less favorable for $J_{12} > 0$ than that for $J_{12} < 0$. Typical spin configurations obtained from numerical calculations are presented in Figs. 7 and 8. It is clear that the skyrmion has opposite helicity between the nearest neighbor layers. Besides the skyrmion lattice phase, the standard magnetic spiral state is stabilized in the low magnetic field region and the spins are fully polarized at high magnetic fields.

Very recently, lattice spin Hamiltonian which reduces to Eq. (5) in the continuum limit by taking the ordering wave vector $q \to 0$ was studied by Hayami [38]. Our phase diagram differs from that in Ref. [38], where more 3-$Q$ states have been identified besides the skyrmion lattice phase. One possible reason for the discrepancy is due to the underlying spin lattice symmetry. In Ref. [38], spins are defined on triangular lattice with $q = \pi/3$, while in the present work, we discretize the continuum model into a square lattice with a fine mesh to minimize the discretization effect. Our method reproduces the known results well when $J_{12} = 0$. The spin lattice generates a potential to orientate the spin modulation wavevector, which can be quite different for the triangular and square lattice for a relatively large $q$. It is known from the previous study [39] that the underlying spin lattice symmetry can have profound consequences on the spin textures that can be realized. In Ref. [38], Hayami restricted to the six in-plane wave vectors $\mathbf{q}_i$ with $|\mathbf{q}_i| = \pi/3$ that are related by six-fold rotation in the simulations. This may also be a source of the discrepancy. In contrast, we have taken all the $q$s permitted by discretization into account in our numerical calculations.

## V. DISCUSSION AND SUMMARY

Here we discuss the dynamics of skyrmion lattice in the presence of a spin Hall torque. When the current is large such that the skyrmion lattice in two layers both depin from impurities and decouple from each other, the two skyrmion lattices move in the opposite direction. If we take

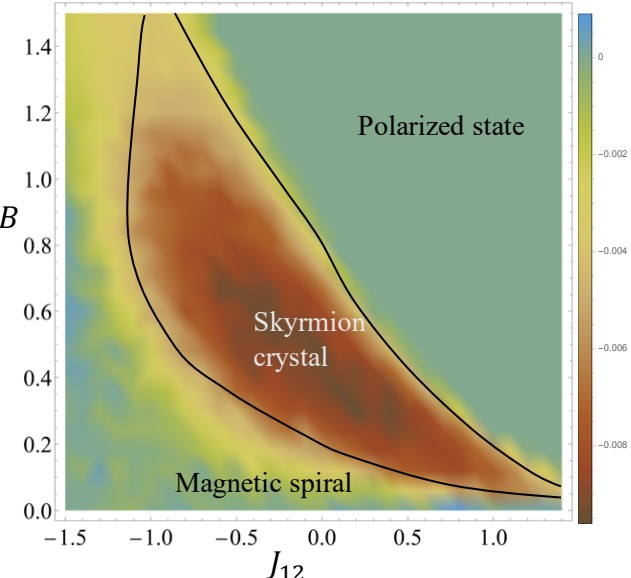

FIG. 6. Phase diagram of the model Eq. (5) in two dimensions (minimal two layers limit). The color represents the skyrmion density. The triangular skyrmion lattice is stabilized in a wide region in the parameter space of $J_{12}$ and $B$. The phase boundary is constructed based on magnetization, skyrmion topological density and their derivatives as a function of $J_{12}$ and $B$. The system size is $40 \times 40$.

skyrmion lattice in one layer as a reference, then effectively the skyrmion lattice in the other layer experiences a periodic potential produced by the reference skyrmion lattice. This produces an oscillating component in the skyrmion lattice velocity with period $T_v \propto a_L/v_{12}$ where $a_L$ is the skyrmion lattice parameter and $v_{12}$ is the relative velocity. In the presence of conduction electrons, the coupling between the skyrmion lattice and conduction electrons generates an oscillating emergent electric field. [40] It is known in the case of Abrikosov vortex lattice, driving the vortex lattice with an ac current in addition to a dc current produces Shapiro steps when the frequency associated with the periodic passing of vortex lattice through pinning sites is commensurate with the ac current frequency. [41, 42] Here similarly phenomenology is expected when the skyrmion lattice is driven through defects. No ac current is required in this case because the relative motion of the two skyrmion lattices already provides an oscillating velocity component. The situation is similar to fractional vortex lattice in multi-component superconductors discussed in Ref. [43], with a notable difference that the Magnus force is dominant over the viscous force for the dynamics of skyrmions.

   We then discuss some open questions for future study. One direct extension of the current work

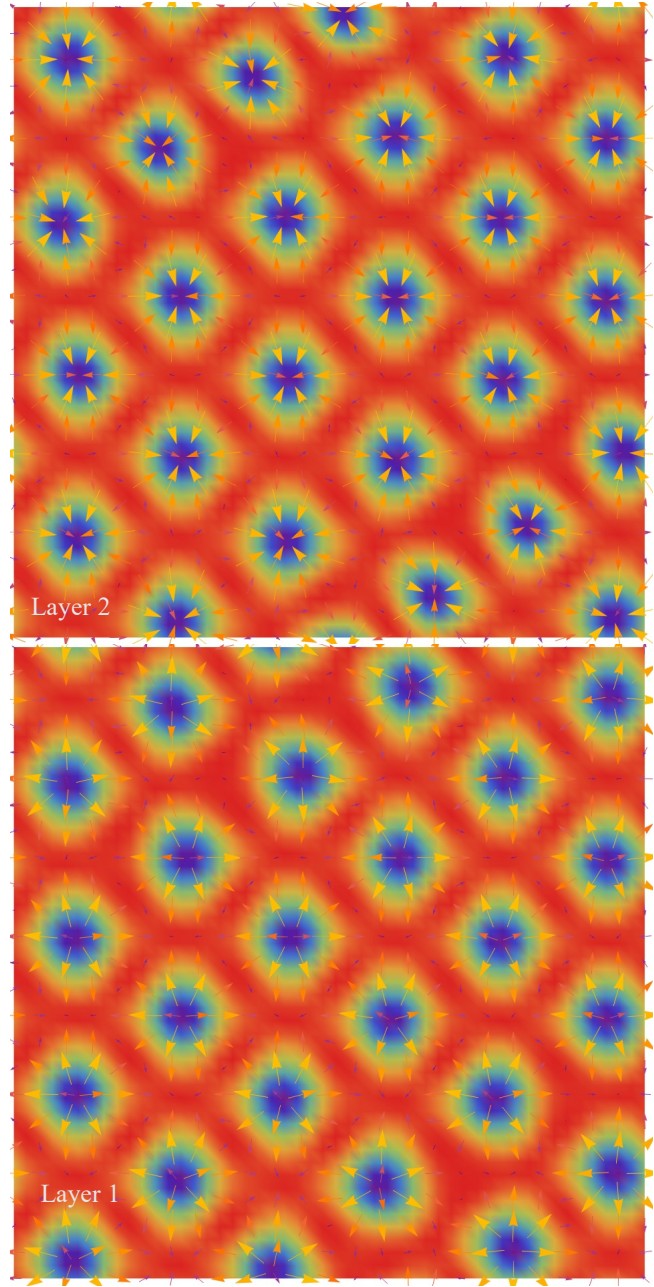

FIG. 7. Skyrmion lattice configuration in the neighboring two layers for $J_{12} = 0.6$ and $B = 0.24$. The skyrmion lattice in one layer is sitting at the interstitial sites of the neighboring layer, and with an opposite helicity. Here color represents out-of-plane component of spin (blue: -1 and red: +1) and arrows denote the in-plane component. The system size is $40 \times 40$.

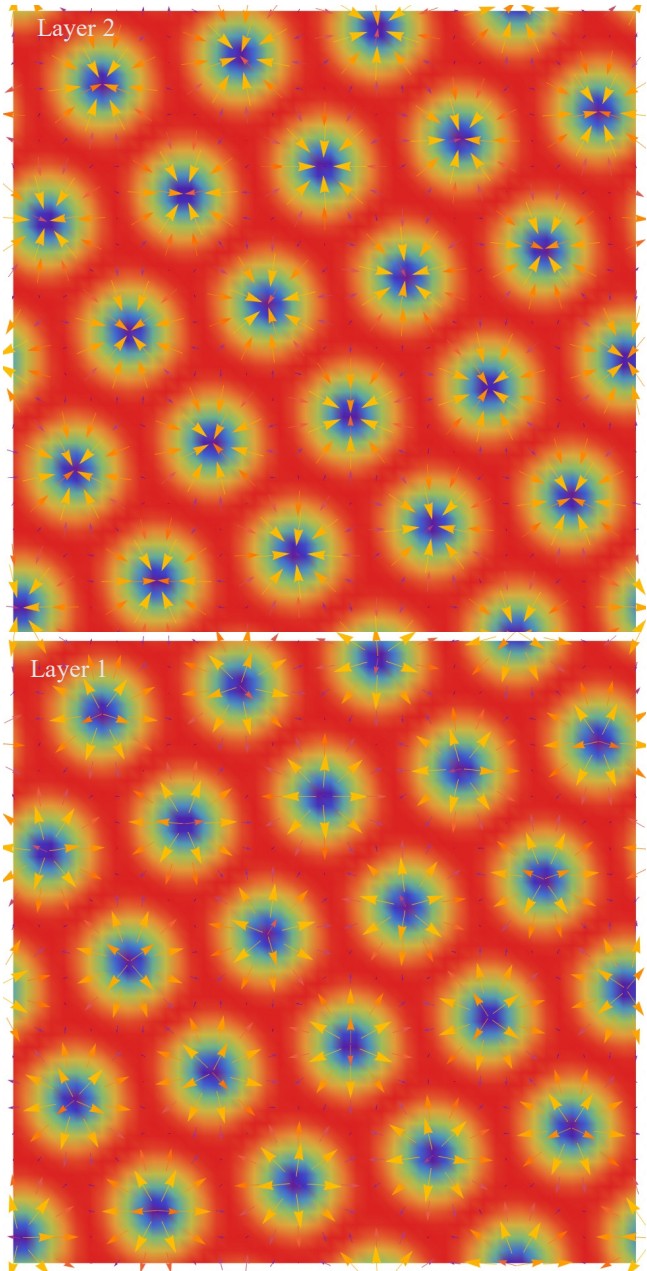

FIG. 8. Skyrmion lattice configuration in the neighboring two layers for $J_{12} = -0.5$ and $B = 0.9$. The skyrmion lattices in the two layers are aligned, and with an opposite helicity. Here color represents out-of-plane component of spin (blue: -1 and red: +1) and arrows denote the in-plane component. The system size is $40 \times 40$.

is the phase diagram in three dimensions. It is known that the magnetic phase diagram in two dimensions and three dimensions differs radically for chiral magnets [5, 6, 44], and we expect the same is also true here. Here we have neglected the interlayer spin orbit coupling. If such a spin orbit coupling is taken into account, there can be an extra twist of spins along the direction perpendicular to layers. This can generate three dimensional skyrmion texture with different skyrmion lattice symmetry. This idea has been validated in three dimensional inversion symmetric magnets with frustrated interlayer couplings [45]. Going beyond the simple spin orbit coupling vectors and local DMI vectors distribution studied here, more complex local DMI vectors distribution are possible depending on the local inversion symmetry breaking pattern, atomic crystal structure and constrained by Moriya's rule [15], which is likely to generate a plethora of spin textures for spins sitting at different sub-lattices.

To summarize, we demonstrate that there exists skyrmion lattice in centrosymmetric magnets with a local inversion symmetry breaking. The lack of local inversion symmetry allows for a local DMI which is sufficient to stabilize the skyrmion spin texture. For the particular crystal structure considered here, the DMI in the nearest neighboring layer has alternating sign and the skyrmion in the neighboring layers has opposite helicity. The dynamics of skyrmions can be quite different when the external drive, such as spin Hall torque, couples to the helicity of the skyrmions. We have determined the equilibrium skyrmion lattice phase diagram and demonstrated the stability of the skyrmion lattice against the interlayer coupling. The skyrmion with alternating helicity in different layers can be detected experimentally using the standard imaging methods, such as neutron scattering spectroscopy, and Lorentz transmission electron microscopy imaging at a surface with a controlled layer termination. The local DMI mechanism identified here is expected to have broader implications in determining spin textures in centrosymmetric magnets, which has largely been overlooked so far. Our results point to a new direction to search for skyrmions in magnetic materials.

## VI.  ACKNOWLEDGEMENTS

This work was carried out under the auspices of the US DOE NNSA under Contract No. 89233218CNA000001 through the LDRD Program. Computer resources for numerical calculations were supported by the Institutional Computing Program at LANL.

Note added– We become aware a recent manuscript [38], which has some overlap with the current work.

------

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
