# Peer review of "Skyrmion lattice in centrosymmetric magnets with local Dzyaloshinsky-Moriya interaction"

_SciPost Physics_

## Round 1 · Referee Report · Anonymous (Referee 1) · 2022-1-17

Report

In a present manuscript, the author considers a bilayer system with the DMI having the opposite sign in each layer. This leads to a stability of coupled Neel skyrmions with inward and outward magnetization rotation. The coupling between the layers is considered to be both FM and AFM. This results in some relative shift of skyrmions in adjacent layers. Moreover, the author considers dynamics of such skyrmions for the spin transfer torque and the spin Hall effect. The current theoretical model is argued to be applied to a material CaBe2Ge2.
In principle, the manuscript is, in most cases, very clear. The topic of this study is of interest for a skyrmionic community. I therefore think that the manuscript is suitable for publication.

In the following, I list some points that should/could be improved and additionally clarified.

1. In the introduction, the phrase “it is always demanding to find a new mechanism to stabilize skyrmions” sounds a bit ambiguous to me. I believe that in any case, the stabilization mechanism boils down either to higher-order derivatives (like for competing exchange interactions leading to frustration) or to linear derivatives of the magnetization with respect to spatial coordinates (DMI, Lifshitz invariants). So, what would be a new mechanism to find?
2. In Fig. 3, would it be not better to express the relative separation of skyrmions in some relative units like the characteristic period of a spiral state? Or alternatively, one could comment how big is this separation with respect to the skyrmions size? What would be the physical reason to have exactly this balanced separation for given parameters?
3. The phrase “The two skyrmions in the skyrmion pair form a bound state with attraction between the two skyrmions” sounds a bit ambiguous. What does it mean attraction? In which sense? Isn't it better to use the word "coupling"? Usually one would use the word ”attraction” trying to investigate the interaction between two such coupled pairs of skyrmions, isn’t?
4. It is a bit hard to understand that for J12<0 skyrmions sit on top of each other and the relative shift is zero. If I consider spins in the skyrmion centers (or at the outskirt), they are parallel in this case, but this is against the AFM coupling. So, why is there no shift?
5. I believe it would be beneficial for a reader to supply some figures with animations. For example, the process of skyrmion decoupling in Fig. 5 would be nice to watch visually. One could see moving skyrmions with the trajectories of their centers.
6. Moreover, I wonder what would be the spin structure after the skyrmions have decoupled in Fig. 5? Are there any oblique spins in the neighboring layer induced by the skyrmions in another layer?
7. What would be the corresponding scenario for J12>0 for the process in Fig. 5?
8. The current manuscript looks very timely, since some similar calculations are claimed to be done in Ref. [38]. However, as the author points out, the results are very different. Would it be possible to reproduce just any result from Ref. [38] to clearly understand where this discrepancy comes from? Otherwise, it sounds as a handwaving that the difference might stem from the spin lattice symmetry. Indeed, if the author of the present manuscript has all numerical codes at hand, why not to reproduce just one result?
9. Could the author comment whether in Eq. (4) periodic boundary conditions are used along all x,y,z, directions? Or is it a bilayer with the free boundaries along z?
10. In Fig. 6, it is not clear why the phase boundaries between different states cannot be plotted explicitly? I.e., the color signifies the density of skyrmions. But it looks like the density of skyrmions is non-zero even in the region of the thermodynamical stability of spirals and within the polarized state. What is the meaning of that?
11. I wonder, would the author predict even more complicated cases when the DMI stabilizes even different types of skyrmions (like skyrmions and anti-skyrmions) in two layers?

---

## Round 1 · Referee Report · Anonymous (Referee 2) · 2022-2-3

Report

I have carefully read this manuscript by Shi-Zeng Lin who studied skyrmions in layered systems where the sign of DMI alternates from layer to layer. He finds that skyrmions are stabilized by the DMI even though the system possesses a global inversion center. This finding contradicts a common narrative in the field of magnetic skyrmions, i.e., that global inversion symmetry should be broken in order to have DMI-stabilized skyrmions. Therefore, I agree that the results presented in this paper are highly interesting as they add a new perspective for materials hosting non-trivial textures such as skyrmions.
As the author of the study also notes in the end of the manuscript, a similar study by Hayami was posted on arXiv roughly one week prior to the submission of this manuscript and, meanwhile, it has been published in PRB. The study by Hayami also considers alternating DMI but focusses on triangular lattices and an in-depth analysis of the zoo of multi-Q states. In turn, the focus of the present study in on the coupling of skyrmions in adjacent layers and their current-driven dynamics, Sec. III, and a demonstration that the DMI stabilizes a skyrmion lattice, Sec. IV.

In the following, let me comment on the manuscript and its contents as I go through the text.
(1) In the introduction, I agree that the honeycomb lattice is an instructive example to start with, even if not directly related to the model compound studied in this manuscript. However, there is a typo because “including such a DMI to the nearest neighbor ferromagnetic Heisenberg model” on a honeycomb lattice would be boring because, as discussed a little further above, the DMI on the nearest neighbor bond vanishes and only arises on the NEXT nearest neighbor bond.
(2) I find it very hard to understand Fig.1b because of the colors and many bonds. Maybe it is better to include a third more schematic panel.
(3) On page 5, the statement that the DMI is non-perturbative seems very nontrivial to me. There is an inversion center, after all. You could cite Hayami now but that would also feel strange if you want to claim that your results are independent and obtained simultaneously. Maybe you just make this more a claim for the moment and for a proof refer to Sec.IV.
(4) Let me note here that you consider the large wavelength limit which is again distinct from Hayami’s paper.
(5) Concerning the required magnetic field B for a polarized background you write that it should be “much larger that |J_{12}| in J_{12}<0” which is the AFM coupling case (there’s a typo in the manuscript in the subscript). Can you give a more precise expression? |J_{12}| is only one part of the energy that has to be outplayed by the B-field.
(6) The distance between the skyrmions in the two layers seems to be determined by the interlayer interaction of the m_z=0 parts. I wonder if you can increase the importance of the core magnetization once the uniaxial anisotropy becomes relevant. Maybe you can comment on this.
(7) More importantly, I wonder how you could determine a finite equilibrium distance for J_12 = 0, i.e., where the layers are completely decoupled. I don’t think that r_m is well-defined at this point and I would expect that there is a discontinuity at J_12=0. Unfortunately, you don’t describe how you obtained this result. But if you started from an initial guess and relaxed the texture, I guess the skyrmions might have gotten trapped in local energy minima. Please verify this point. I think it would also be interesting to show the potential energy landscape of the skyrmion interaction.
(8) What happens outside the region of stability? Does the background become unstable? Does one of the skyrmions decay by shrinking? If so, can you refer to previous studies? If not, what happened instead?
(9) I don’t understand how the halo in the adjacent layer makes sense. How do you unwind the vortex? Does it smoothly unwind on a very short length scale? Or is there a true discontinuous vortex core? Since you do not indicate your numerical lattice constant in Fig2 but only in the other panels and it is very very large, did you check if this is maybe a discretization effect that vanishes in the continuum limit?
(10) It is maybe interesting to note that omnipresent dipolar interactions will break the symmetry between the skyrmions in the two layers.
(11) The current-driven dynamics are interesting but you should compare them to analytical results combined with the numerically calculated potential landscape (see my comment #7).
(12) Whether you get inertia effects or not probably depends on your definition of inertia. If you stop the current in the simulation in Fig.5b after some time before the dissociation, the skyrmions will keep moving for some time until the internal energy has dissipated and they are back at their equilibrium distance. This is inertia effect (automation) can be interpreted as a mass induced by J_12.
(13) In the phase diagram, is the phase shift smooth as function on J_12 as you cross the decoupled case?
(14) I agree that your study of the phase diagram is more rigorous as the restricted model that Hayami used. But your numerical discretization is very poor and the phase diagram is very noisy, in particular given that there is no temperature or other source of noise. Can you please clarify this point. Please, also explain how you obtained the phase diagram. Is this Monte Carlo or direct energy minimization of trial states? The latter could give much less noisy results with less numerical effort once you know which phases with which approximate symmetries play a role.
(15) Is the non-triangular lattice in Fig.7 a numerical artefact or a real feature?
(16) For more than 2 layers, do the skyrmions stack rather ABABAB or ABCABC? A projected magnetization map could also be very helpful for identification in transmission experiments!

In summary, the present work by Lin predicts the stabilization of meta-stable bound skyrmion pairs as well as thermodynamically stable skyrmion lattices in centrosymmetric magnets with alternating DM interaction from layer to layer. Moreover, the dynamics under applied currents are discussed. The prediction of a skyrmion phase was almost simultaneously published by Hayami in PRB. In particular, given the huge amount of comments and criticism, I cannot recommend the publication of this manuscript in SciPost. Moreover, since this manuscript appeared after the one by Hayami in PRB and only adds minor advances, I don’t believe that its publication in the more impactful journal SciPost would be justified.

---

## Editorial Decision

awaiting_resubmission